# Research on the Interactive Relationship of Spatial Expansion between Estuarine and Coastal Port Cities

**Zeyang Li, Weixin Luan \*, Zhenchao Zhang and Min Su**

School of Maritime Economics and Management, Dalian Maritime University, Dalian 116026, China
* Correspondence: weixinl@dlmu.edu.cn

**Abstract:** In both developed and developing countries, port-city relationships have always attracted much attention. However, in the port–city interface, views differ as to whether the port drives the city or vice versa. The combination of remote sensing data and geospatial big data (point of interest) has provided a favorable solution. Taking the typical estuarine and coastal port cities in China's coastal zone as an example, this study examines the following contents based on the port–city interface: the formation age of urban built-up areas and port areas on both sides of the port–city boundary; interaction between port and urban built-up areas; and the distribution of urban functional areas outside the port. Results show that the degree of spatial integration in estuarine port cities is higher than that of coastal port cities and that in the past 30 years, the expansion of ports has led to the expansion of cities. This expansion is port- and sea-oriented, and the expansion direction of the port city is consistent. On the port–city interface, the estuarine and coastal port cities form different urban regional structure modes. Aside from enriching literature on the port–city relationship, this study provides a reference for the spatial planning and transformation of ports and cities in the future.

**Keywords:** spatial interaction; port–city interface; estuarine port-city; coastal port-city





## 1. Introduction

Port cities have always been one of the key research objects in urban geography, urban and rural planning, economics, sociology, urban management, and other related fields [1–11]. This is because, as a part of the city, the port has always been a land node for exchanges between different countries and cultures. Ports not only contribute to the urban economy but also occupy certain interdependent urban spaces [12]. As the node of transportation, the port continuously transports materials and energy for the city. The city uses the port to attract more enterprises and talents, thereby enhancing the city's competitiveness. The relationship between the port and the city is inseparable [13]. In most economically developed countries and regions—such as London and Rotterdam in Europe, New York and Los Angeles in North America, and Dubai, Yokohama, Busan, Singapore, Hong Kong, and Shanghai in Asia—cities and ports have been integrated [7,14] and have become important international hubs and economic centers. For China and other developing countries, foreign trade is the main economic orientation [15]. Therefore, port cities have become an important gateway to promote national and regional economic globalization and foreign exchanges [16]. With the improvement of the global logistics system and the integrated development of the supply chain, the importance of some ports and cities has become more apparent. Both need to make more contributions to economic development in a limited space. The spatial relationship between ports and cities has become an significant factor affecting the development of ports, cities, and coastal areas [17–19]. The port is also the core advantage of an open city, and the spatial relationship between the port and the city is also the key factor affecting the sustainable development of the port city. Therefore, the spatial relationship and evolution process of ports and cities in the past must be systematically studied. Especially in the context of port–city integration, it

is important to understand the development of the port city and the urban spatial layout on the port–city interface. It can also be used as an important basis for port-city spatial planning and sustainable development in the future.

China's coastal port cities have a long history. Depending on locations, they can be divided into estuarine port cities and coastal port cities [20]. Since the 1980s, in the context of economic globalization, China's coastal port system and scale have been expanding. According to statistics, in 1978, the cargo throughput of China's coastal ports was less than 300 million tons, and the container throughput was only 100,000 TEUs. In terms of container throughput in 1982, there were no shortlisted Chinese ports among the top 100 global ports. However, by the end of 2020, the cargo throughput of Chinese coastal ports reached 14.55 billion tons, which was 48.5-fold more than that in 1978, and the container throughput reached 260 million TEUs. Both the cargo and container throughputs are ranked first globally. The cargo throughput for eight Chinese ports ranks among the top 10 globally, while the container throughput of seven ports ranks among the top 10 globally. With the increase in port size, five major port groups have been formed in the Chinese mainland coastal zone, namely the Bohai port group, the Yangtze River Delta port group, the southeast coastal port group, the Pearl River Delta port group, and the southwest coastal port group. Among them, port cities such as Dalian, Qingdao, Shanghai, Ningbo, Guangzhou, and Zhanjiang are the core hubs of the above five port clusters, and coastal city clusters have gradually developed into port-city clusters. Therefore, research on the spatial interaction between ports and cities in China's coastal zone is more typical.

Based on the above information, taking estuarine port cities (Shanghai, Ningbo, and Guangzhou) and coastal port cities (Dalian, Qingdao, and Zhanjiang) as examples (Figure 1), this study examines the spatial interaction between estuarine and coastal port cities on the port–city interface (Figure 2) by using remote sensing data and data from point of interest. First, the formation age of urban built-up areas and port areas on both sides of the port–city boundary is considered. Second, the spatial evolution process of the built-up area and the port area on both sides of the port–city boundary, the interaction between the port and the city, and the direction of spatial expansion are analyzed. Finally, the distribution of urban functional areas on the port–city interface is quantitatively identified. The research results can provide a reference for the spatial planning and sustainable development of ports and cities in the future. Additionally, in the field of management, previous research on the spatial relationship between port and city mainly focused on the macroeconomic perspective and less on the change of port and city patches. The research paradigm combining remote sensing and big data used in this study can provide new ideas for port and urban management research in the future.

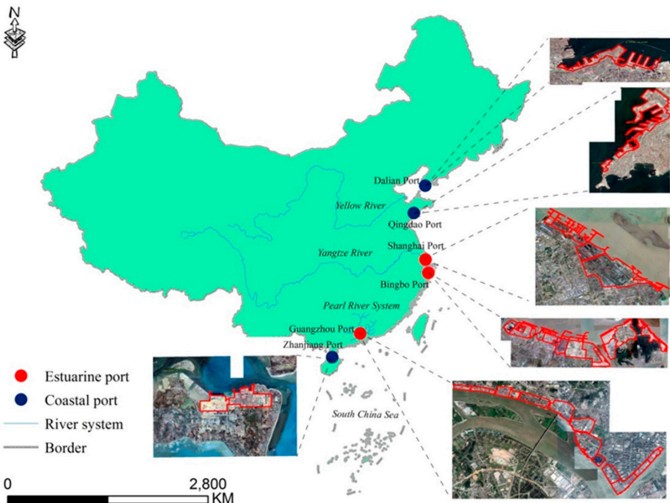

**Figure 1.** Location of study area (source: drawn by author).

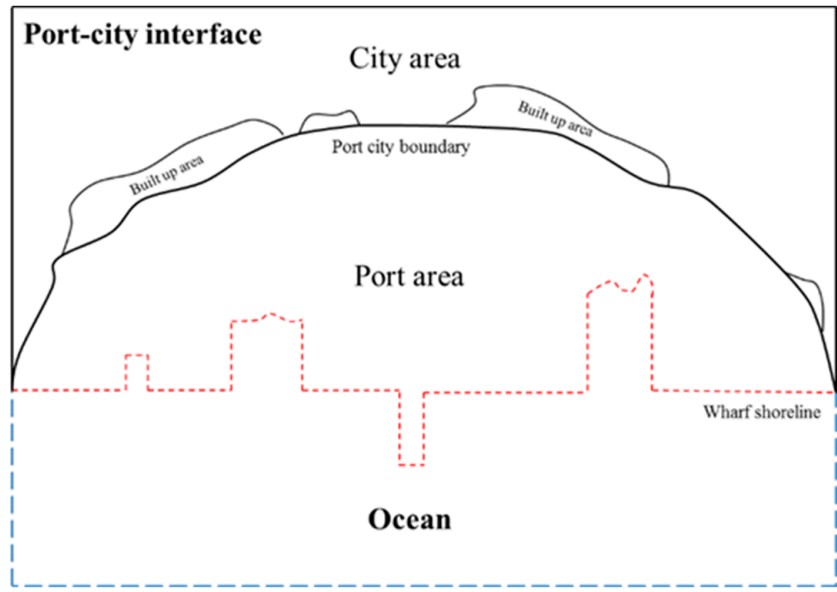

**Figure 2.** Schematic diagram of port-city interface (source: drawn by author).

## 2. Literature Review

### 2.1. Theory of Port–City Relationship

The relationship between the port and the city was first put forward by British geographers. When they studied the perspective of port facility expansion, they summarized the temporal and spatial evolution law of Britain's main seaports [21]. Then, scholars elaborated the general theory and development law of port-city interaction from the perspectives of port–city interaction, port-city growth model, port-city life cycle, and port-city integration [22–25]. In terms of port–city interaction, Daamen and Vries et al. proposed a variety of forms of port–city interaction, including spatial location, industrial correlation, economy, and culture [26]. Chinese scholar Liu was the first to elaborate on the theory of port–city interaction [27]. He believed that the interaction between port and city usually refers to the process of mutual causality and interactive development in the evolution of the relationship between the two. The theory of port–city interaction refers to the leading and driving role of the port in the process of urban development and the port's promotion and service role when the economic and social activities of the city gather to a certain extent. The two go hand in hand, which completely reflects the internal mechanism of port–city interaction. In terms of the mode of port-city growth, British scholars first divided the port growth process into the original development stage, edge wharf expansion stage, port area renovation stage, general linear wharf development stage, and professional wharf development stage [28]. After the 20th century, it mainly expounds the growth theory of the port city from the perspective of spatial evolution and economic correlation [29]. Chinese scholar Wu discussed the law of economic development of seaport cities and reached a new conclusion that the growth model of seaport cities has different phased effects [30]. Xu pointed out that the important driving force for the growth of port cities comes from ports. The growth mode of port cities corresponding to the growth force of port cities can be divided into four types, namely the initial connection of port cities, correlation between port cities, agglomeration effect of port cities, and self-growth effect of cities [31]. In terms of the port-city life cycle theory, Vigarie established the concept of port-city life cycle [32]. Chinese scholar Chen believed that the relationship between the port and the city comes in different phases and used the port-city life cycle theory to analyze the development stage of the port city [33]. This is the first time that the life cycle theory of the port city was concluded in China. In terms of port-city integration, in 1934, German scholar Gauz used Weber's industrial location theory to draw an important view of port-city integration based on the relationship between the port and the hinterland [21]. Bird and Olivierô et al. put

forward the concept of "Big port and Big city", showing that the two are integrated after the highest degree of coordinated development [34]. Chinese scholars Xu et al. divided the extension of port-city integration into four levels, namely the integration of the port and related urban project construction, the integration of the port and related urban spatial layout, the integration of the port and other transportation modes of the city, and the integration of the port and the strategic objectives of the city [35]. Taking Liaoning Province of China as an example, Liu et al. studied the interactive relationship between port cities and established the sustainable development framework of port cities [36].

## 2.2. Spatial Relationship between Port and City

Ports and cities affect each other in space, culture, society, system, and economy. Hayuth first put forward the concept of "port-city interface" in 1982 and tried to confirm its scope of change from the triple system of port–city connection (space, economy, and ecology) [37]. Hoyle believed that as an "area in transition", the port–city interface is the most sensitive and controversial place. It is the result of the joint action of several factors and requires careful and appropriate planning methods [24]. Hoyle also pointed out that the dynamic mechanism causing the change of the port–city interface includes: (1) migration of port activities from the traditional core area of port-city to deep-water area and places with more land and sea space; (2) that industries relying on ports no longer rely on labor and will migrate to other places outside the city; (3) increase in waterfront development activities; and (4) influence and control of technological progress, legal provisions, economy and politics, environment, and other factors [38]. Since the 21st century, the rise of seaports in developing countries, especially in Asia, has made scholars focus on the spatial changes of global seaports and their cities and not only of Western seaports [5]. Gleave earlier studied the impact of port activities on urban space in Africa and believed that the spatial correlation between port activities, central business, and industrial areas is common in tropical Africa [39]. Lee et al. studied the spatial evolution of Asian hub-port cities and proposed a coordination relationship model taking Asian hub-port cities as an example [18]. Wiegmans and Louw found that in the port of Amsterdam, the expansion speed of port space slowed down, while the city gradually expanded towards the port and that the relationship between port and city had entered a new stage [40]. Van den Berghe and others believed that the port–city interface is a geometric relationship through which the heterogeneity of participants, assets, and structures are coupled with each other. They constructed an analytical framework taking the contemporary geometric relationship between the biological departments of Amsterdam and Ghent as the starting point. In their analysis, they revealed how different coupling mechanisms formed the specific development trajectory of port cities, laying a foundation for the coupling mechanism in the future [41]. Gurzhiy et al. studied the port–city integration of Shanghai, Rotterdam, and Hamburg and believed that space, IT strategic planning based on digital network, and land and sea traffic management would help maintain the sustainability of the port-city economy [19].

As seen in the existing literature, ocean and coastal zone management is an all-round and multi-level research topic [42–44]. As an important part of the coastal zone, seaports, and their cities, the interaction between the two is of great importance to coastal management and even the land and marine coordinated development strategy of China [43]. The author believes that the interaction between port and city has the following three impacts on coastal management and land and sea planning: first, the impact on coastal zone spatial planning should be considered. The spatial planning and management of ocean and coastal is a gradual process. Montoya Rojas GA et al. [45] studied the dynamic relationship between the biogeographic environment of the coastal and the port waterfront cities, formulated the sustainability principle through multi standard analysis, and provided contributions to the urban planning and management in the coastal environment. However, the study only conducted qualitative analysis from the perspective of policies and documents and did not conduct quantitative research. From the perspective of economic management,

Huang et al. [46] extended Alvin Toffler's "three waves" concept into a classification framework that described three stages and six types of port-city developments. Using the series of steps in constructing the framework, the purpose of obtaining maximum social benefits and appropriate risk management was thus achieved. Different from the abovementioned research, the present research quantitatively analyzes the interaction between various types of coastal ports and cities. Starting from the port–city interface, this study defines the sequence of port-city spatial expansion, describes the spatial interaction and expansion direction of the port city, and analyzes the urban regional structure mode on the port–city interface, and on the basis of previous studies, the theory of coastal spatial planning and management is supplemented. Second, we consider the impact on the dynamic changes of coastal environment. The development of ports and cities, especially the increase of reclamation activities, will inevitably pose a threat to the coastal environment. As Yue said [47], in the past decade, China's annual average reclamation area was about 100 square kilometers. Land reclamation provides space guarantee for industrialization and urbanization, representing the driving force of China's rapid economic growth. At the same time, the rapid expansion of land reclamation has also brought problems of excessive scale and extensive marine utilization, resulting in the loss of a large number of coastal shoals, wetlands, and natural coastlines as well as the continuous deterioration of the marine ecological environment [48–50]. Ding et al. [51] also showed that the increase of reclamation activities in China's coastal areas had a significant negative impact on habitat fragmentation. Third, the impact of port and urban development on land and sea transportation system needs to be mentioned [52]. The interactive development of ports and cities has promoted the improvement of land and sea transportation system and the construction of transportation facilities. The research of Li et al. [20] has proven that the opening of new ports has promoted the construction of port-city railways and highways and also promoted the formation of new urban built-up areas.

## 3. Materials and Methods

### 3.1. Data Sources

The spatial data of coastal ports used in this paper is from a Google Earth satellite remote sensing snapshot; the shooting time was from June to September 2020. Each port and port area has a snapshot, and a total of 29 satellite remote sensing snapshots of coastal port areas were obtained. In this paper, only six typical ports are selected as the research objects. Urban land-use data are from multi-period RS monitoring dataset of land-use and -cover changes in China (CNLUCC), which consist of vector data from 1990 to 2020 (from the Institute of Geographical Sciences and Nature Resources Research, Beijing, China). The main attributes include ID, patch length, patch area, and land-use type. The land-use types are divided into six categories, namely cultivated land, forest land, grassland, water area, urban and rural, industrial and mining, residential land, and unused land. Among them, port and urban spatial data are mainly obtained from urban and rural, industrial and mining, and residential land. Points of interest (POI) data were obtained from the Baidu map. The study mainly uses the POI data of estuarine port cities (Shanghai, Ningbo, Guangzhou) and coastal port cities (Dalian, Qingdao, Zhanjiang) in 2020.

### 3.2. Data Processing and Research Unit Division

The port boundary data were obtained through the remote sensing snapshot of the Google Earth satellite and the spatial planning map of each port. Using the spatial analysis function of ArcGIS software, the port boundary and the CNLUCC of different periods from 1990 to 2020 were analyzed spatially to obtain the spatial evolution data of the port from 1990 to 2020. Due to the duplication and redundancy of POI data, they cannot be used directly, and the original data needed to be reclassified. Referring to the latest edition of urban land classification and planning and construction land standards in 2011 and following the principles of universality and consistency, this study divides the POI data into six categories: residential land, public service land, commercial service land, industrial

land, transportation land, and green space and square land (Figure 3). Since POI is a point without area and volume, in reality, the physical area of POI in the research unit has an important impact on the unit's land-use type. Therefore, through remote sensing images and online data query, the average building area or floor area of various POIs is roughly determined [53], and various POI points are weighted with reference to the public awareness ranking of POI proposed by Zhao et al. [54], with a weight score range of 1–100. For features with large building or floor area and high public awareness, the weight score will be correspondingly high, such as railway stations, parks, factories, and the like. The specific weight determination results are as follows: transportation land (100), green space and square land (90), industrial land (80), public service land (50), residential land (30), and commercial service land (20). According to various POI weights, the POI data set required for the study were finally generated.

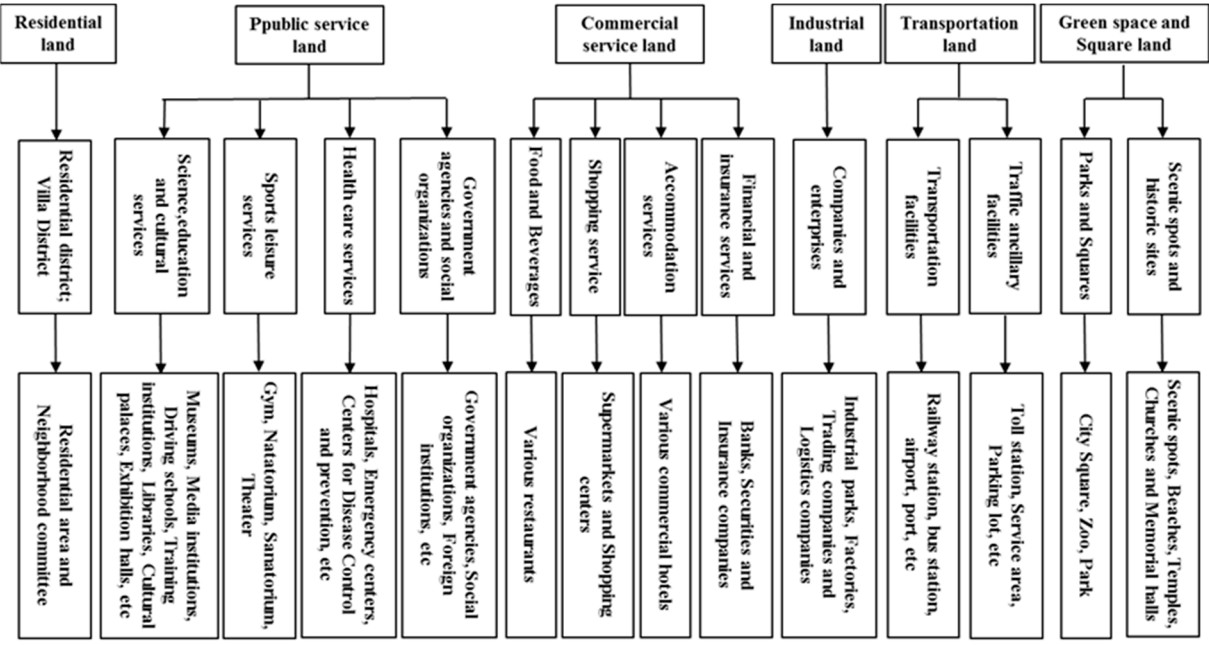

**Figure 3.** Reclassification of POI data (source: drawn by author).

As an infrastructure transportation facility, the impact of the port on its surrounding area is similar to that of other transportation facilities (such as railway stations, bus stations, and subway stations). Scholars at home and abroad generally use the influence radius of 400–800 m to delimit the influence scope of transportation facilities [55,56]. However, due to the large area and wide radial range of the port based on field investigations, this study delimits the 2 km distance around the port as the radial range of the port on the port–city interface. Additionally, referring to the identification methods of urban functional areas by many scholars [57,58], the 300 × 300 m grid is defined within the radial range of each port as the basic research unit. The number of research units within the radial range of each port is as follows: Shanghai (2397), Ningbo (2932), Guangzhou (5174), Dalian (2381), Qingdao (1430), and Zhanjiang (1879).

*3.3. Methods*

3.3.1. Graphical Induction

Remote sensing images provide a better record of the spatial evolution process of geographical objects. The spatial expansion process of a harbor must be reproduced via RS and geographic information system technology. Based on the interface between the port and the city, this research constructs the concept of the boundary between the two and tries to determine the formation order of the spots on both sides of the boundary by determining their formation age. In this way, we can directly observe the evolution process of port and

city patterns and finally identify the mutual driving relationship between port and city in the port-city spatial expansion process. Therefore, this study uses graphic induction to study the spatial expansion process of port and city. This method has been proven effective in the past research on the spatial expansion law of estuarine and coastal ports [20]. The main experimental steps are as follows: first, the vector layers of estuarine and coastal ports and cities in 1990, 2000, 2010, and 2020 were obtained by using multi-period RS monitoring dataset of land-use and -cover changes in China (CNLUCC). Second, using spatial analysis tools from the Arcgis 10.5 software, the layers of ports and cities in different years were superimposed to obtain the new patches of ports and cities in different years. Finally, the length of port–city boundary in different years was obtained using superposition analysis function and ranging tool to facilitate subsequent research and analysis.

3.3.2. Identification of Urban Functional Areas Based on POI Data

Referring to the research of Chi et al. [53], this study identifies the type of urban land around the port by constructing index frequency density (F) and category ratio (C). The calculation formula is as follows:

$$F_i = \frac{n_i}{N_i} \ (i = 1, 2, \ldots, 6) \tag{1}$$

$$C_i = \frac{F_i}{\sum\limits_{i=1}^{6} F_i} \times 100\%, \ i = 1, 2, \ldots, 6 \tag{2}$$

where, $i$ is POI type; $n_i$ is the number of $i$th types of POI in the unit; $N_i$ is the total number of $i$th types of POI; $F_i$ indicates the frequency density of the $i$th type of POI in the total number of this type of POI; $C_i$ is the ratio of the frequency density of the $i$th type of POI to the frequency density of all types of POI in the unit.

## 4. The Formation Age of Urban Built-Up Area on the Port–City Interface

Table 1 shows the length of urban built-up areas in different years on the boundary of estuarine and coastal port cities (among them, "proportion" refers to the length percentage of the construction land patches formed in different years on the boundary between the port and the city of the total length of the boundary). Generally speaking, on the estuary port–city interface, the average proportion of port–city boundary length that has formed the urban built-up area is 62.76%, whereas the average proportion of not-formed urban built-up areas is 37.24%. Among them, the largest proportion of the length of the port–city boundary that has formed the urban built-up area is Shanghai, accounting for 75.87%, followed by Guangzhou and Ningbo, accounting for 68.56% and 46%, respectively. On the coastal port–city interface, the average proportion of port–city boundary length that has formed urban built-up area is 56.76%, whereas the average proportion of not-formed urban built-up areas is 43.24%. Among them, the largest proportion of the length of the port–city boundary that has formed the urban built-up area is Zhanjiang, accounting for 84.51%; followed by Dalian, accounting for 73%; and finally Qingdao, accounting for 40.82%. The above results show that the degree of spatial integration of estuarine port cities is higher than that of coastal port cities. Among the estuarine port cities, Shanghai port city has gradually integrated and realized port-city spatial integration. The degree of integration of Guangzhou port city is second only to Shanghai, and Ningbo still has large port–city boundary resources to be developed. Among the coastal port cities, the port–city boundary resources of Zhanjiang and Dalian have been basically developed, while nearly 60% of the port–city boundary resources of Qingdao have not formed urban built-up areas, so the development of port–city boundary resources has high potential.

**Table 1.** Patch length of built-up area formed in different years on the port–city boundary.

| Type | City | Length of Port–City Boundary (km) | Urban Age | Patch Length of Built-Up Area (km) | Proportion | Type | City | Length of Port–City Boundary (km) | Urban Age | Patch Length of Built-Up Area (km) | Proportion |
|---|---|---|---|---|---|---|---|---|---|---|---|
| Estuarine port city | Shanghai | 78.7 | 1990 | 17.81 | 22.63% | Coastal Port city | Dalian | 68.89 | 1990 | 11.25 | 16.33% |
| | | | 2000 | 10.69 | 13.58% | | | | 2000 | 4.8 | 6.97% |
| | | | 2010 | 7.6 | 9.66% | | | | 2010 | 1.16 | 1.68% |
| | | | 2020 | 23.61 | 30.00% | | | | 2020 | 33.08 | 48.02% |
| | | Total | - | 59.71 | 75.87% | | | Total | - | 50.29 | 73.00% |
| | Ningbo | 90.65 | 1990 | 15.29 | 18.96% | | Qingdao | 68.01 | 1990 | 10.95 | 16.10% |
| | | | 2000 | 0.23 | 0.29% | | | | 2000 | 0.87 | 1.28% |
| | | | 2010 | 5.72 | 7.09% | | | | 2010 | 2.24 | 3.29% |
| | | | 2020 | 20.45 | 22.56% | | | | 2020 | 13.7 | 20.14% |
| | | Total | - | 41.69 | 46% | | | Total | - | 27.76 | 40.82% |
| | Guangzhou | 82.44 | 1990 | 11.9 | 8.81% | | Zhanjiang | 53.57 | 1990 | 13.56 | 25.31% |
| | | | 2000 | 11.25 | 8.33% | | | | 2000 | 1.3 | 2.43% |
| | | | 2010 | 15.5 | 11.48% | | | | 2010 | 0 | 0.00% |
| | | | 2020 | 17.97 | 5.90% | | | | 2020 | 15.2 | 28.37% |
| | | Total | - | 56.62 | 68.56% | | | Total | - | 30.06 | 84.51% |
| Total | | 251.79 | - | 158.02 | 62.76% | Total | | 190.47 | - | 108.11 | 56.76% |

Note: The "proportion" refers to the length percentage of the construction land patches formed in different years on the boundary between the port and the city of the total length of the boundary.

To further analyze the time sequence of the formation of ports and urban built-up areas and clarify the spatial interaction between ports and cities, this study uses statistics on the patch length of urban built-up areas formed earlier or later than ports on the estuarine and coastal port–city boundary (Table 2). Overall, on estuarine port–city interface, the average patch length of urban built-up area formed earlier than the port accounts for 38.12%, and the average patch length formed later than the port accounts for 61.88%. Among them, the largest proportion of urban built-up areas formed later than the port is in Ningbo, accounting for 73.66%, followed by Shanghai and Guangzhou, accounting for 54.13% and 53.12%, respectively. On the coastal port–city interface, the average patch length of urban built-up area formed earlier than the port accounts for 24.22%, and the average patch length formed later than the port accounts for 75.78%. Among them, the largest proportion of urban built-up areas formed later than the port is Qingdao, accounting for 79.33; followed by Dalian, accounting for 75.02%; and finally Zhanjiang, accounting for 72.26%. The above results show that whether it is estuarine or coastal port city, the urban patch formed later than the port accounts for a large proportion of the port–city interface, indicating that the port expansion always drives the urban expansion in the process of port-city formation. However, the proportion of urban built-up areas formed earlier than ports in estuarine port cities is larger than that of coastal port cities. This is because the estuarine port cities include inland river ports formed in the early stage. Since the Chinese population has the habit of residing near water, before the formation of ports, there was an urban layout around river systems, which is in line with the regional characteristics of Chinese cities. This is different from the estuarine port city, where there are only coastal ports in the coastal port city, and before the formation of the port, the urban area is small. The formation of the port drives the further expansion of the city, and the driving effect of the port on the city is more apparent than that of the estuarine port city.

**Table 2.** Patch length of built-up area formed earlier/later than the port on the port–city boundary.

| Type | City | Length of Port–City Boundary (km) | Formed Earlier than the Port | | Formed Later than the Port | |
|---|---|---|---|---|---|---|
| | | | Length (km) | Proportion | Length (km) | Proportion |
| Estuarine port city | Shanghai | 78.7 | 36.1 | 45.87% | 42.6 | 54.13% |
| | Ningbo | 90.65 | 21.24 | 26.34% | 69.41 | 73.66% |
| | Guangzhou | 82.44 | 38.65 | 46.88% | 43.79 | 53.12% |
| Total | | 251.79 | 95.99 | 38.12% | 155.8 | 61.88% |
| Coastal port city | Dalian | 68.89 | 17.21 | 24.98% | 51.68 | 75.02% |
| | Qingdao | 68.01 | 14.06 | 20.67% | 53.95 | 79.33% |
| | Zhanjiang | 53.57 | 14.86 | 27.74% | 38.71 | 72.26% |
| Total | | 190.47 | 46.13 | 24.22% | 144.34 | 75.78% |

Note: In the statistical process, the part of the port–city boundary that does not form a built-up area is defined as the part formed later than the port.

In sum, there are differences between estuarine and coastal port cities in the process of spatial formation. Therefore, based on the research results of Li et al. and Table 1, this study finds that for the estuarine port cities, in the estuarine port stage, the port and the city integrate and develop, and the port drives its periphery to form an early rising urban built-up area. At this time, the port is the main driving force for the expansion of urban built-up area. In the coexistence stage of the estuarine and coastal ports, with the expansion of the port, new urban built-up areas are gradually formed on the port–city interface, but this proportion is smaller than that in the previous stage. In the deep-water port stage, the port will be transferred to the deep-water bay area or island far away from the city. The formation of a new port area will drive the formation of new urban built-up areas in its periphery. Even though 37.24% of the port–city boundary has not formed built-up areas, with the continuous promotion of port–city integration, the remaining parts still have great potential to become new urban built-up areas.

For the coastal port city, in the coastal port stage, the port area is the urban area, but the scope of the urban built-up area is small. The port is the original driving force for the formation of the city, and the port is also the city's original function. With the diversified development of port areas and functions (reflected in two levels: first, in terms of function: transportation or portal → commerce → industry; second, in terms of region: there is not only the expansion of a single port region but also the increase of the number of port regions on the coastline), the urban built-up areas serving port functions and urban functions gradually grow on the port–city boundary, and the expansion of ports drives the expansion of urban built-up areas. In the coastal port stage on the edge of the city, since the port and the city are not separated by natural factors such as rivers and mountains in space, the expansion of the port and the formation of new port areas will still drive the formation of new urban built-up areas. In the deep-water port stage, with the expansion of the port city, the port and the urban built-up area will eventually be connected, and the port city presents an integrated pattern in space. Although 43.24% of the port–city boundary lines have not formed urban built-up areas, the development potential is higher than that of estuarine port cities.

## 5. The Driving Effect of the Port on the Urban Built-Up Area of the Port–City Interface

The driving effect of port on the expansion of urban built-up area on the port–city interface shows the spatial interaction between port and city. Figure 4 (a. Shanghai, b. Ningbo, c. Guangzhou) and Figure 5 (a. Dalian, b. Qingdao, c. Zhanjiang) show the spatial changes of estuarine and coastal port cities. Overall, the average proportion of urban built-up areas formed before 1990 in estuarine port cities is only 17.87%, while the average proportion from 1990 to 2020 is 82.13%. The average proportion of urban built-up areas formed before 1990 in coastal port cities is only 18.78%, while the average proportion from 1990 to 2020 is 81.22%. It can be seen that after the formation of the port, it drives the rapid expansion of urban built-up areas. The periphery of the new port area will also gradually form a new urban built-up area, and the port city will be gradually connected in space. Therefore, in both estuarine and coastal port cities, the expansion of ports drives the expansion of urban built-up areas, and this expansion is port-oriented and sea-oriented, and the expansion direction of port cities is consistent.

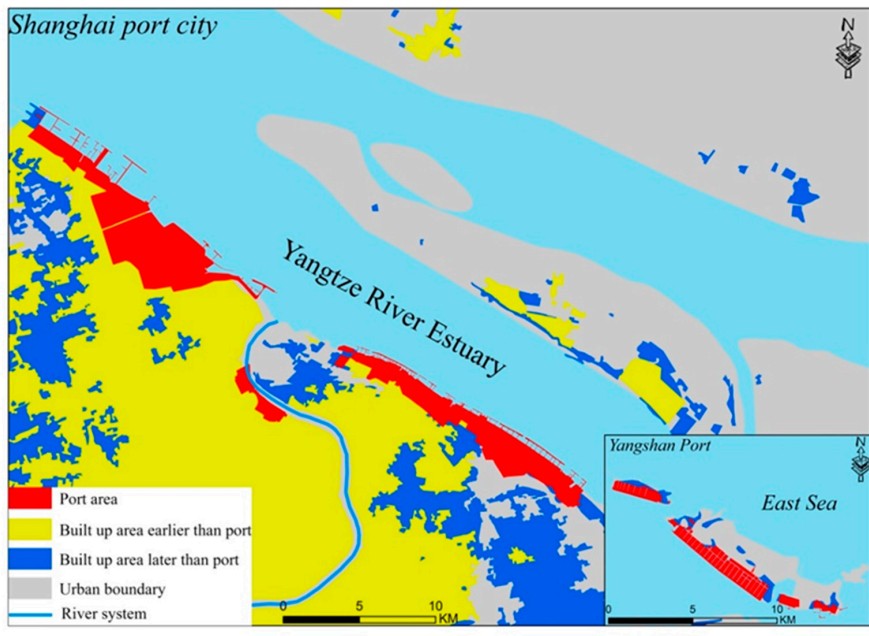

**a**. Spatial change of Shanghai port–city interface.

**Figure 4.** *Cont.*

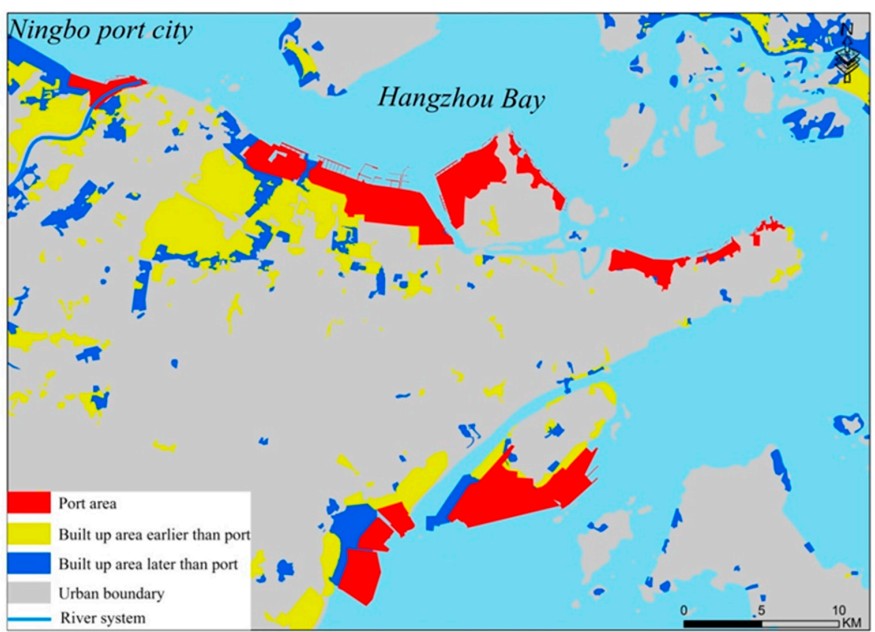

**b**. Spatial change of Ningbo port–city interface.

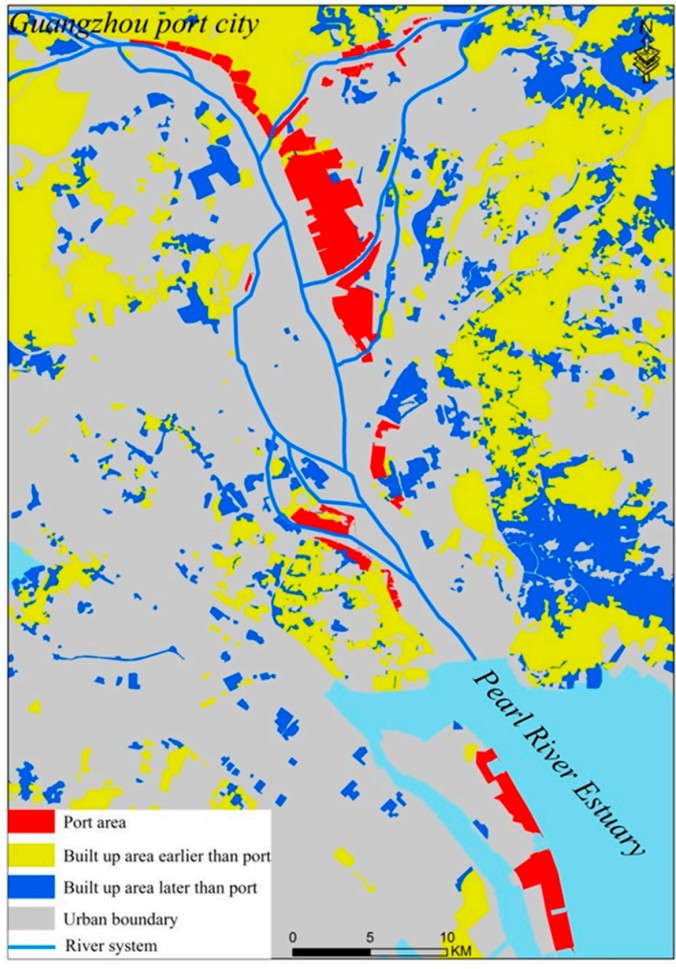

**c**. Spatial change of Guangzhou port–city interface

**Figure 4.** Spatial change of estuary port–city interface (source: drawn by author).

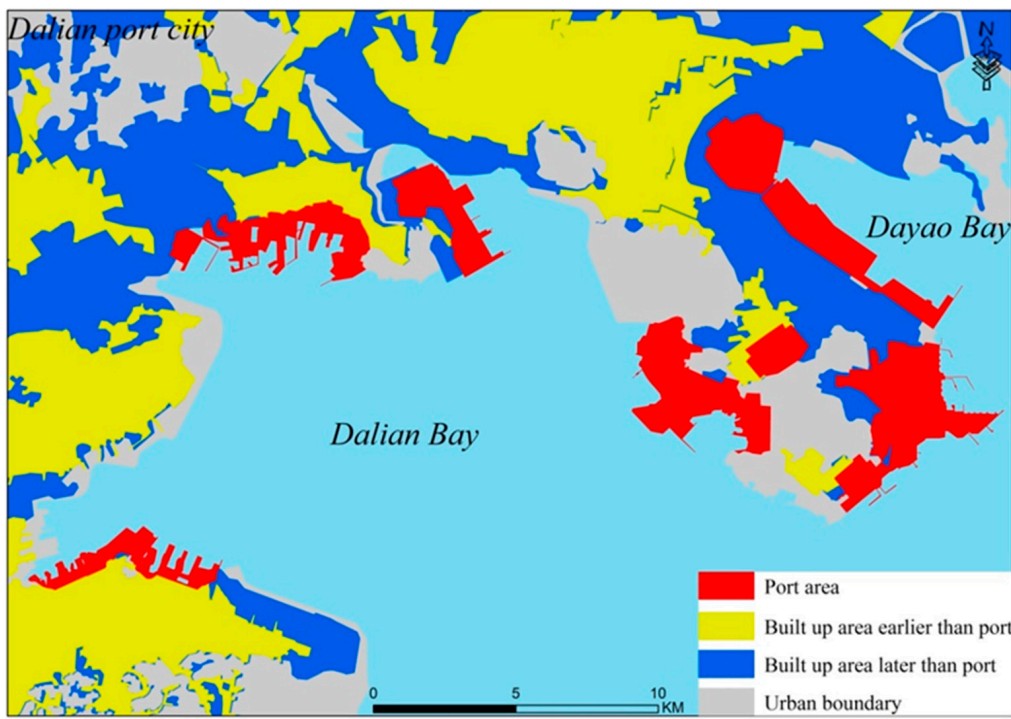

**a**. Spatial change of Dalian port–city interface.

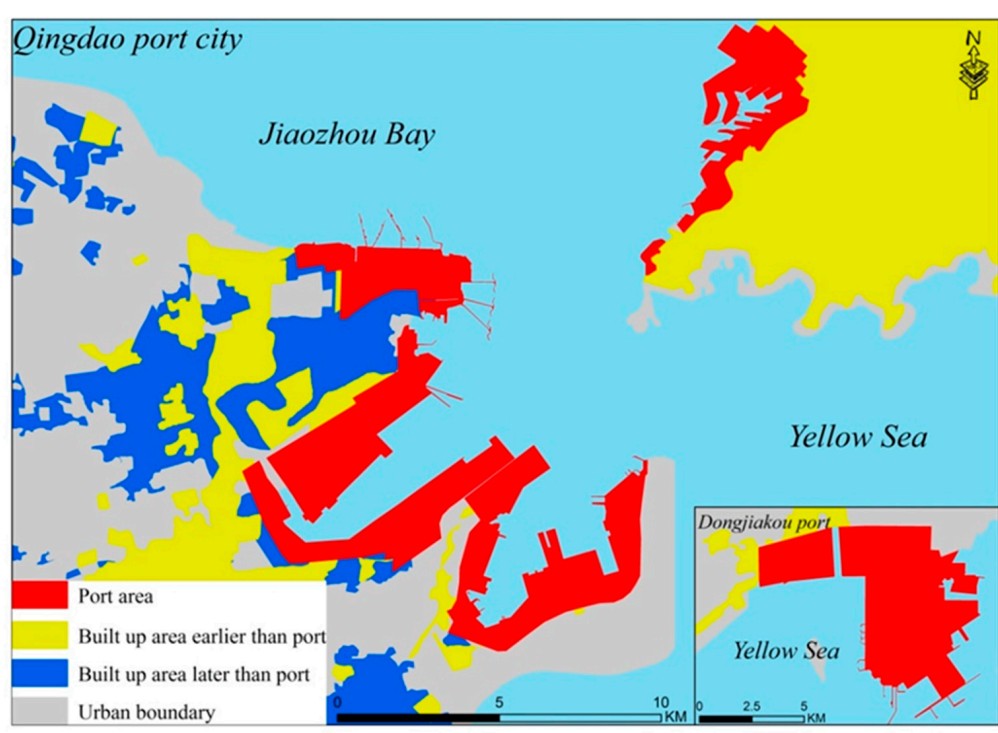

**b**. Spatial change of Qingdao port–city interface.

**Figure 5.** *Cont.*

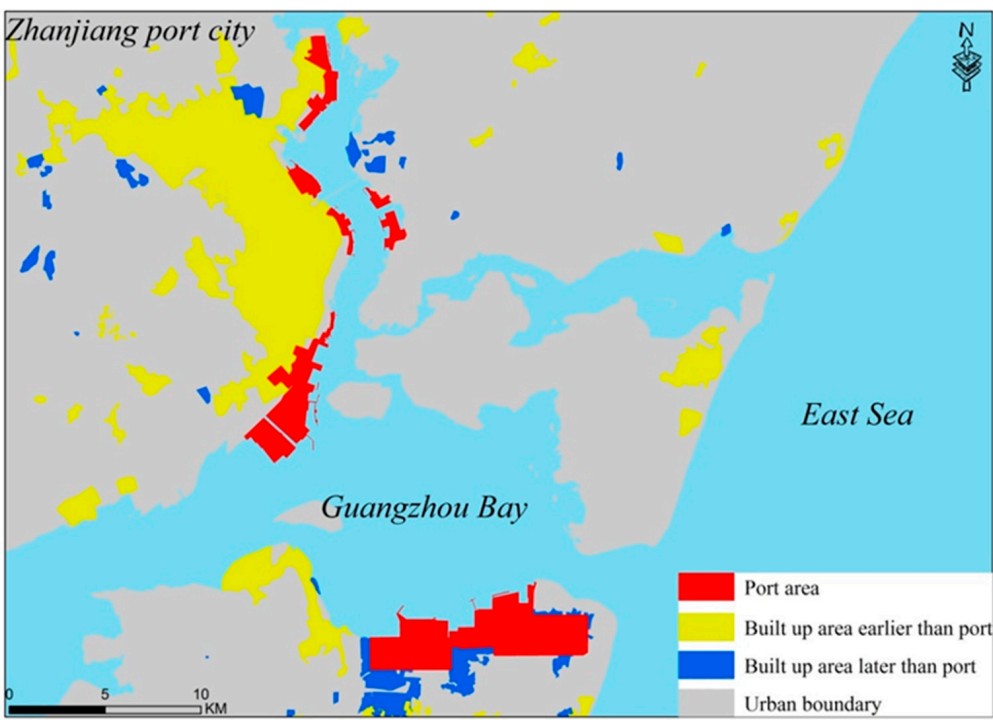

**c**. Spatial change of Zhanjiang port–city interface.

**Figure 5.** Spatial change of coastal port–city interface (source: drawn by author).

Based on the research results of Li et al. [20], specifically, in the development process of the estuarine port city, the port gradually expands to the sea along the downstream of the river, and then develops into a deep-water port. The expansion direction of urban built-up areas has the characteristics of port and sea, and the expansion direction of port and city tends to be the same. This is because estuarine port cities have rivers entering the sea, and the terrain at the river entrance is flat, which is suitable for the formation of urban built-up areas. At the same time, the formation of ports has a gravitational effect on the formation of new urban built-up areas, and the expansion of port cities shows convergence. Table 3 shows the area changes of estuarine and coastal ports and urban built-up areas. Shanghai port mainly extends eastward from the west bank of Huangpu River along the estuary of the Yangtze River, and the urban built-up area also extends from the vicinity of Huangpu River and Wusong River to the east coast. Among them, ports and cities grew the fastest from 2010 to 2020, with an increase of 19.93 km$^2$ and 1053.78 km$^2$, respectively. Ningbo port was formed at the estuary of Yongjiang River, Yuyao River, and Fenghua River, then expanded along Hangzhou Bay and finally to the East China Sea. The urban built-up area also expanded from the upstream of the estuary of Yongjiang River, Yuyao River, and Fenghua River to the northeast coastline. Among them, ports and cities grew the fastest from 2010 to 2020, with an increase of 43.71 km$^2$ and 2372.7 km$^2$, respectively. Guangzhou port mainly expands southward along the Pearl River, and the urban built-up area also expands to the southeast coastal zone. Among them, the port grew the fastest from 2010 to 2020, with 43.71 km$^2$, while the city grew the fastest from 2000 to 2010, with 530.8 km$^2$.

In the development process of coastal port cities, the port mainly expands to the deep-water coast or bay area far away from the city and gradually develops into a deep-water port. The expansion direction of urban built-up area is port-oriented; that is, as long as a new port area is formed, a new urban built-up area will be formed on the port–city interface, and the expansion direction of port city is the same. Table 3 shows the area changes of estuarine and coastal ports and urban built-up areas. Dalian port mainly expands from the south bank of Dalian Bay to the north bank and then transfers to the coast of the Yellow Sea. With the formation of the new port area, the urban built-up area transfers to the new coastal

port. From 2010 to 2020, ports and cities grew the fastest, i.e., 22.98 km$^2$ and 633.58 km$^2$, respectively. Qingdao port city mainly expands from north to south along the coast of the Yellow Sea, and the urban built-up area also expands towards the port. From 2010 to 2020, the port and city grew the fastest, i.e., 45.48 km$^2$ and 229.49 km$^2$, respectively. Zhanjiang port city mainly expands from north to south along the west bank of Guangzhou Bay, and the urban built-up area also expands to the port. From 2010 to 2020, the port and city grew the fastest, i.e., 41.44 km$^2$ and 81.31 km$^2$, respectively.

**Table 3.** Area of estuarine and coastal port cities.

| | | 1990 | | 2000 | | 2010 | | 2020 | |
|---|---|---|---|---|---|---|---|---|---|
| **Type** | **City** | **Port Area (km$^2$)** | **Built-Up Area (km$^2$)** | **Port Area (km$^2$)** | **Built-Up Area (km$^2$)** | **Port Area (km$^2$)** | **Built-Up Area (km$^2$)** | **Port Area (km$^2$)** | **Built-Up Area (km$^2$)** |
| Estuarine port city | Shanghai | 19.67 | 614.24 | 27.42 | 889.93 | 37.82 | 1333.82 | 57.75 | 2387.6 |
| | Ningbo | 7.16 | 1106.72 | 8.54 | 1524.76 | 17.45 | 3707.8 | 67.3 | 6080.5 |
| | Guangzhou | 2.79 | 290.92 | 5.62 | 469.88 | 19.33 | 1000.68 | 63.04 | 1245.89 |
| Coastal port city | Dalian | 1.73 | 723.97 | 6.11 | 912.32 | 6.48 | 1008.71 | 29.46 | 1642.29 |
| | Qingdao | 9.43 | 565.11 | 12.16 | 674.16 | 16.7 | 864.52 | 62.18 | 1094.01 |
| | Zhanjiang | 3.49 | 241.71 | 3.66 | 289.45 | 3.87 | 312.18 | 45.31 | 393.49 |

## 6. Identification of Urban Functional Areas on the Port–City Interface

This study uses POI to identify the urban functional area of the port–city interface. We calculated the frequency density (F) and category ratio (C) of each grid unit according to Formulas (1) and (2) and selected the POI type with the highest proportion as the main land type of the research unit, namely urban functional area. To explore the change characteristics of urban functional areas at different types of port–city interfaces, taking the port–city boundary as the starting point, we counted the number and proportion of urban functional areas within 0–0.5 km, 0.5–1 km, 1–1.5 km, and 1.5–2 km of the periphery, respectively. The results are shown in Table 4.

### 6.1. Distribution of Urban Functional Areas on the Estuarine Port–City Interface

Figures 6–8 show the distribution of urban functional areas on the estuarine port–city interface. It can be seen that the urban functional areas on the port–city interface of Shanghai, Ningbo, and Guangzhou are mainly concentrated in the periphery of the old port area, which was mainly formed in the stage of estuarine port. These areas are also the earliest built-up areas of the city. However, the density of urban functional areas around the new port area is small, and some have not yet formed built-up areas, which belong to urban areas later than the port in time. Combined with Table 4, Figures 6–8, and the actual investigation, the author draws the urban regional structure model diagram of the estuarine port–city interface. As shown in Figure 9, within 0–0.5 km outside the estuarine port, it is mainly dominated by industrial functional areas, supplemented by green space and square functional areas, and inlaid with some residential functional areas and commercial service functional areas. This part of the area is mainly formed in the estuary port stage of the port, and port industries and enterprises mainly gather in this area to reduce transportation costs. Additionally, since the estuarine port was originally formed along the river, it is easy to place the green grassland and square land layout in this area. Within the range of 0.5–1 km, it is mainly residential functional areas to facilitate the commuting of port industrial and enterprise personnel. Accordingly, some transportation functional areas and commercial service functional areas are arranged. Within the range of 1–1.5 km, it is dominated by the public services' functional areas, supported by commercial service functional areas, transportation functional areas, residential functional areas, and green space and square functional areas. The type of functional areas within 1.5–2 km is relatively simple, mainly commercial service functional areas matched with residential functional areas and transportation functional areas.

**Table 4.** Number and proportion of urban functional areas in the port–city interface.

| Type | City | Type of Functional Area | Total | 0–0.5 km | | 0.5–1 km | | 1–1.5 km | | 1.5–2 km | |
|---|---|---|---|---|---|---|---|---|---|---|---|
| | | | | Quantity | Proportion | Quantity | Proportion | Quantity | Proportion | Quantity | Proportion |
| Estuarine port city | Shanghai | Transportation | 74 | 14 | 18.92% | 17 | 22.97% | 18 | 24.32% | 25 | 33.78% |
| | | Residential | 171 | 20 | 11.70% | 44 | 25.73% | 55 | 32.16% | 52 | 30.41% |
| | | Commercial service | 141 | 20 | 14.18% | 30 | 21.28% | 39 | 27.66% | 52 | 36.88% |
| | | Industrial | 192 | 95 | 49.48% | 48 | 25.00% | 42 | 21.88% | 45 | 23.44% |
| | | Public service | 129 | 28 | 21.71% | 29 | 22.48% | 40 | 31.01% | 32 | 24.81% |
| | | Green space and square | 27 | 9 | 33.33% | 6 | 22.22% | 9 | 33.33% | 3 | 11.11% |
| | Ningbo | Transportation | 15 | 1 | 6.67% | 2 | 13.33% | 7 | 46.67% | 5 | 33.33% |
| | | Residential | 46 | 9 | 19.57% | 9 | 19.57% | 12 | 26.09% | 16 | 34.78% |
| | | Commercial service | 140 | 15 | 10.71% | 25 | 17.86% | 42 | 30.00% | 58 | 41.43% |
| | | Industrial | 317 | 81 | 25.55% | 88 | 27.76% | 74 | 23.34% | 74 | 23.34% |
| | | Public service | 75 | 7 | 9.33% | 14 | 18.67% | 34 | 45.33% | 20 | 26.67% |
| | | Green space and square | 14 | 5 | 35.71% | 5 | 35.71% | 1 | 7.14% | 3 | 21.43% |
| | Guangzhou | Transportation | 90 | 14 | 15.56% | 20 | 22.22% | 25 | 27.78% | 31 | 34.44% |
| | | Residential | 106 | 20 | 18.87% | 25 | 23.58% | 22 | 20.75% | 39 | 36.79% |
| | | Commercial service | 204 | 42 | 20.59% | 45 | 22.06% | 40 | 19.61% | 77 | 37.75% |
| | | Industrial | 385 | 124 | 32.21% | 78 | 20.26% | 78 | 20.26% | 105 | 27.27% |
| | | Public service | 145 | 28 | 19.31% | 27 | 18.62% | 45 | 31.03% | 45 | 31.03% |
| | | Green space and square | 5 | 3 | 60.00% | 1 | 20.00% | 1 | 20.00% | 1 | 20.00% |
| Coastal port city | Dalian | Transportation | 46 | 2 | 4.35% | 19 | 41.30% | 8 | 17.39% | 17 | 36.96% |
| | | Residential | 118 | 11 | 9.32% | 18 | 15.25% | 31 | 26.27% | 58 | 49.15% |
| | | Commercial service | 128 | 24 | 18.75% | 29 | 22.66% | 27 | 21.09% | 48 | 37.50% |
| | | Industrial | 217 | 44 | 20.28% | 59 | 27.19% | 58 | 26.73% | 54 | 24.88% |
| | | Public service | 108 | 20 | 18.52% | 14 | 12.96% | 42 | 38.89% | 32 | 29.63% |
| | | Green space and square | 22 | 1 | 4.55% | 6 | 27.27% | 11 | 50.00% | 4 | 18.18% |
| | Qingdao | Transportation | 42 | 10 | 23.81% | 13 | 30.95% | 7 | 16.67% | 12 | 28.57% |
| | | Residential | 55 | 11 | 20.00% | 13 | 23.64% | 7 | 12.73% | 24 | 43.64% |
| | | Commercial service | 61 | 15 | 24.59% | 8 | 13.11% | 12 | 19.67% | 26 | 42.62% |
| | | Industrial | 35 | 8 | 22.86% | 7 | 20.00% | 12 | 34.29% | 8 | 22.86% |
| | | Public service | 37 | 6 | 16.22% | 8 | 21.62% | 13 | 35.14% | 10 | 27.03% |
| | | Green space and square | 38 | 7 | 18.42% | 11 | 28.95% | 14 | 36.84% | 6 | 15.79% |

**Table 4.** *Cont.*

| Type | City | Type of Functional Area | Total | 0–0.5 km | | 0.5–1 km | | 1–1.5 km | | 1.5–2 km | |
|------|------|-------------------------|-------|----------|----------|----------|----------|----------|----------|----------|----------|
| | | | | Quantity | Proportion | Quantity | Proportion | Quantity | Proportion | Quantity | Proportion |
| | Zhanjiang | Transportation | 40 | 9 | 22.50% | 4 | 10.00% | 11 | 27.50% | 16 | 40.00% |
| | | Residential | 50 | 5 | 10.00% | 11 | 22.00% | 13 | 26.00% | 21 | 42.00% |
| | | Commercial service | 66 | 9 | 13.64% | 15 | 22.73% | 16 | 24.24% | 26 | 39.39% |
| | | Industrial | 70 | 24 | 34.29% | 20 | 28.57% | 23 | 32.86% | 3 | 4.29% |
| | | Public service | 80 | 13 | 16.25% | 27 | 33.75% | 22 | 27.50% | 18 | 22.50% |
| | | Green space and square | 11 | 1 | 9.09% | 6 | 54.55% | 3 | 27.27% | 1 | 9.09% |

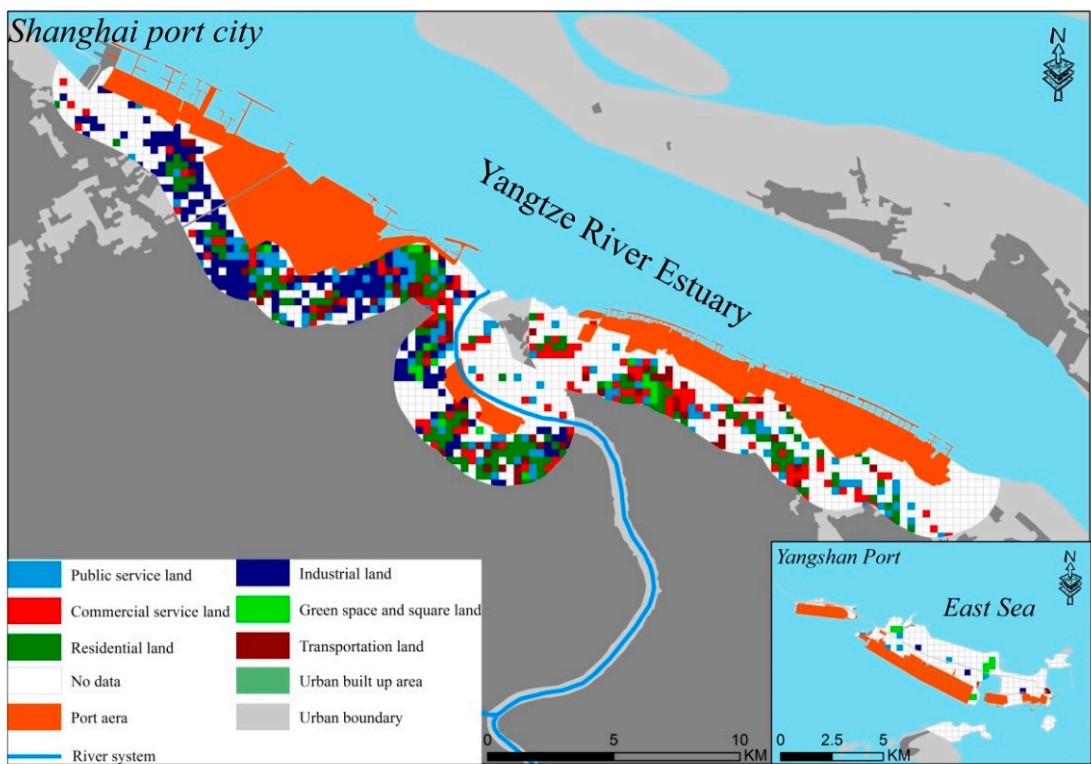

**Figure 6.** Distribution of urban functional areas on the Shanghai port–city interface (source: drawn by author).

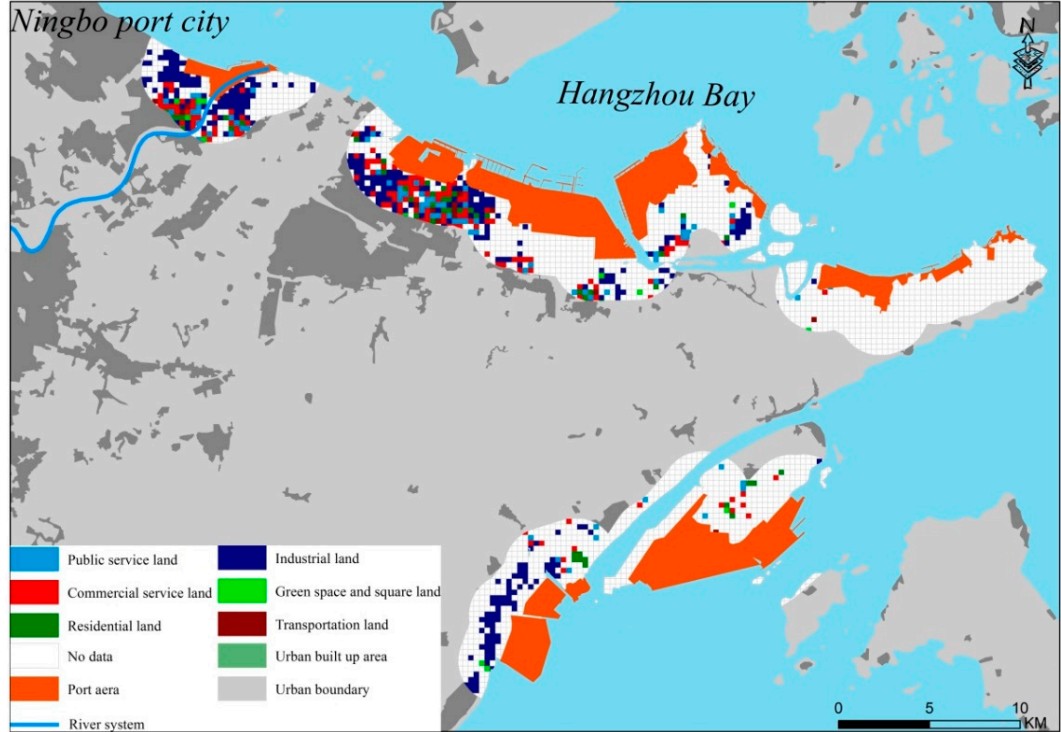

**Figure 7.** Distribution of urban functional areas on the Ningbo port–city interface (source: drawn by author).

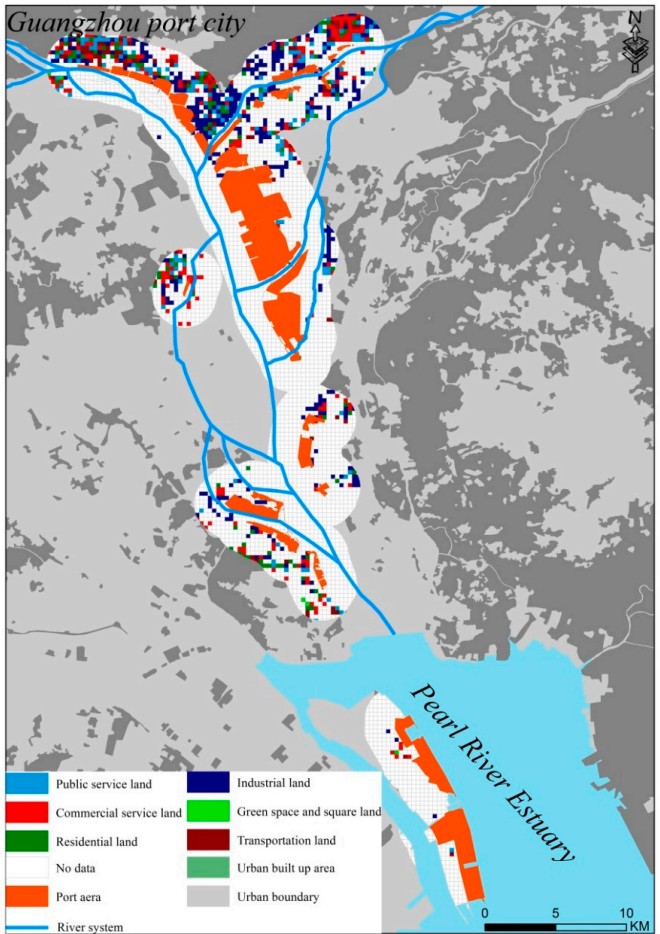

**Figure 8.** Distribution of urban functional areas on the Guangzhou port–city interface (source: drawn by author).

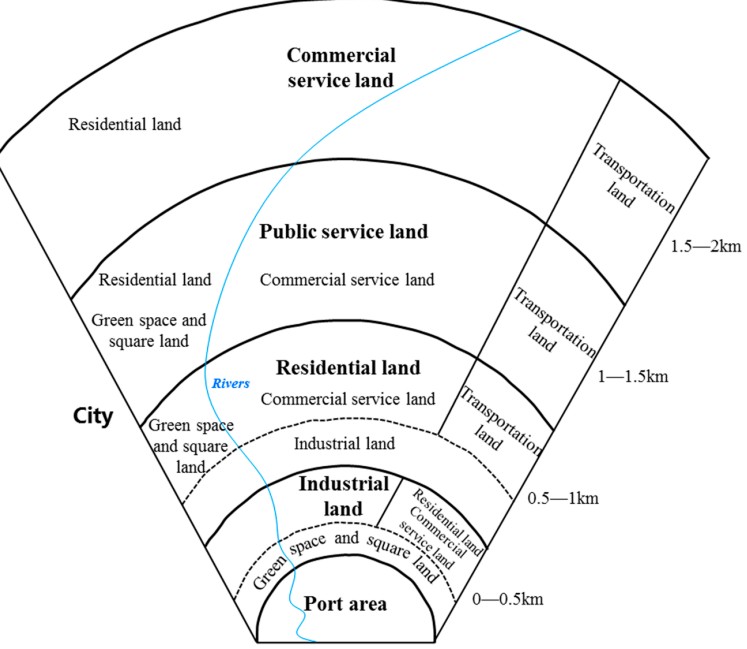

**Figure 9.** The urban regional structure model diagram of the estuarine port–city interface (source: drawn by author).

*6.2. Distribution of Urban Functional Areas on the Coastal Port–City Interface*

Figures 10–12 show the distribution of urban functional areas at the interface of coastal port city. Similar to estuarine ports, the urban functional areas on the port–city interface of Dalian, Qingdao, and Zhanjiang are mainly distributed in the periphery of the old port area, which is mainly formed in the coastal port stage of port evolution. This is because in the early stage of coastal port city, the city prospered with port. Therefore, the earliest built-up area of the city was formed outside the old port area, and the density of urban functional areas was large. As the port evolves into a coastal port and deep-water port far away from the city, new urban functional areas begin to form in the periphery of the new port area, but the density is small, and some do not yet form built-up areas, which belong to the urban areas later than the port in time. Combined with Table 4, Figures 10–12, and the actual investigation, the author draws the urban regional structure model diagram of the coastal port–city interface, as shown in Figure 13. In the periphery of coastal port, the transportation function areas run through the whole port–city interface. Within the range of 0–0.5 km, it is mainly industrial functional areas, which are embedded in transportation functional areas, public service functional areas, and commercial service functional areas. This part of urban functional area is mainly formed in the coastal port stage of port evolution. As a logistics distribution center, the port will inevitably drive the layout of urban industrial enterprises around it, which is also the basic condition for urban development. Within the range of 0.5–1 km, it is mainly green space and square functional areas, supplemented by industrial functional areas and residential functional areas. This is because the coastal port will face the problem of urban waterfront reconstruction in the deep-water port stage [59]. With the reduction of the port function of the old port area, various urban tourist attractions will be formed around it, and then, green space and square functional area will be formed. Consistent with the estuarine port city, the public service functional areas are mainly arranged within the range of 1–1.5 km. The type of functional areas within 1.5–2 km are mainly commercial service functional areas matched with residential functional areas and transportation functional areas.

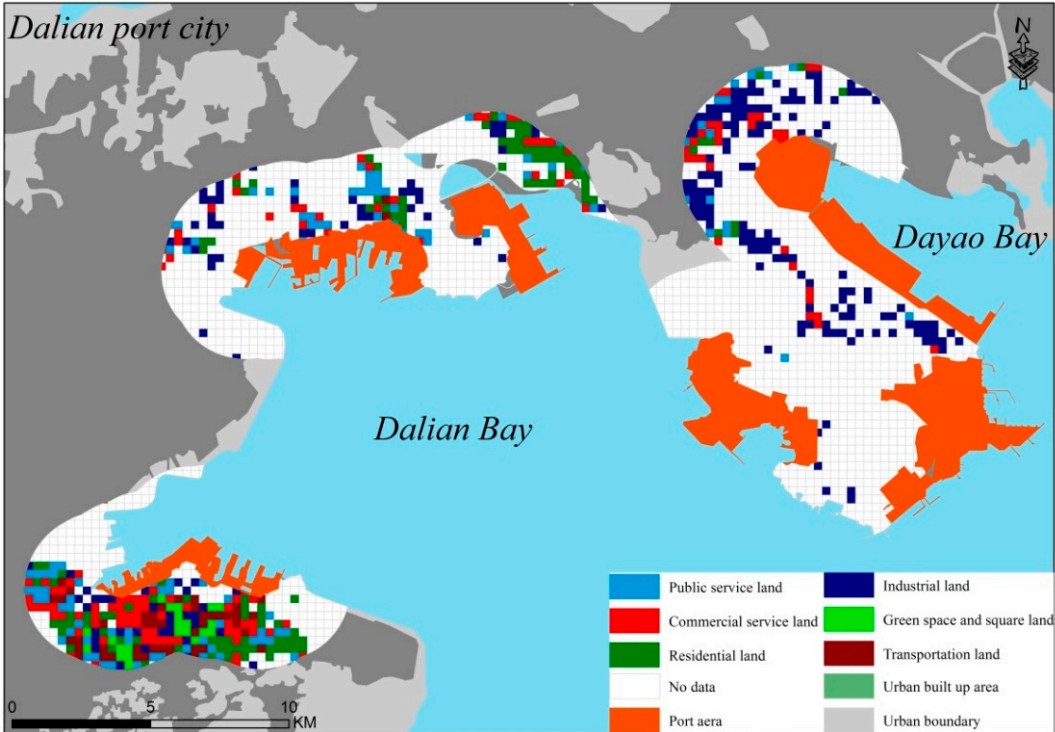

**Figure 10.** Distribution of urban functional areas on the Dalian port–city interface (source: drawn by author).

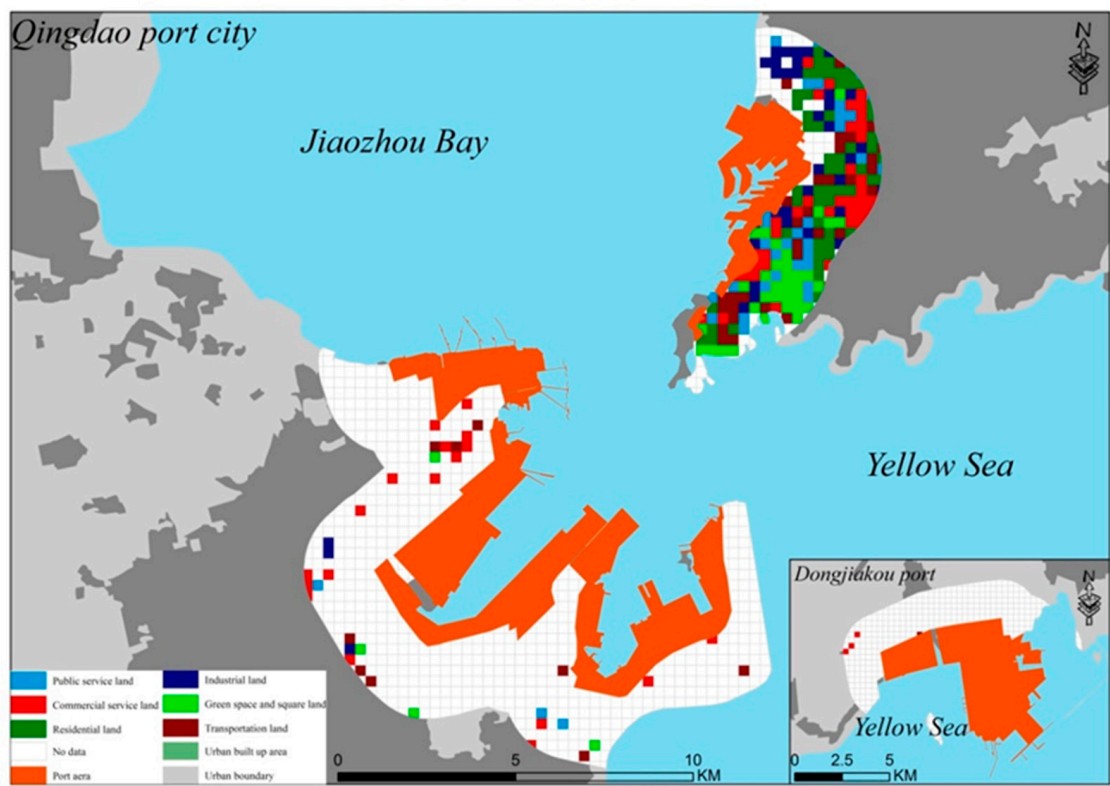

**Figure 11.** Distribution of urban functional areas on the Qingdao port–city interface (source: drawn by author).

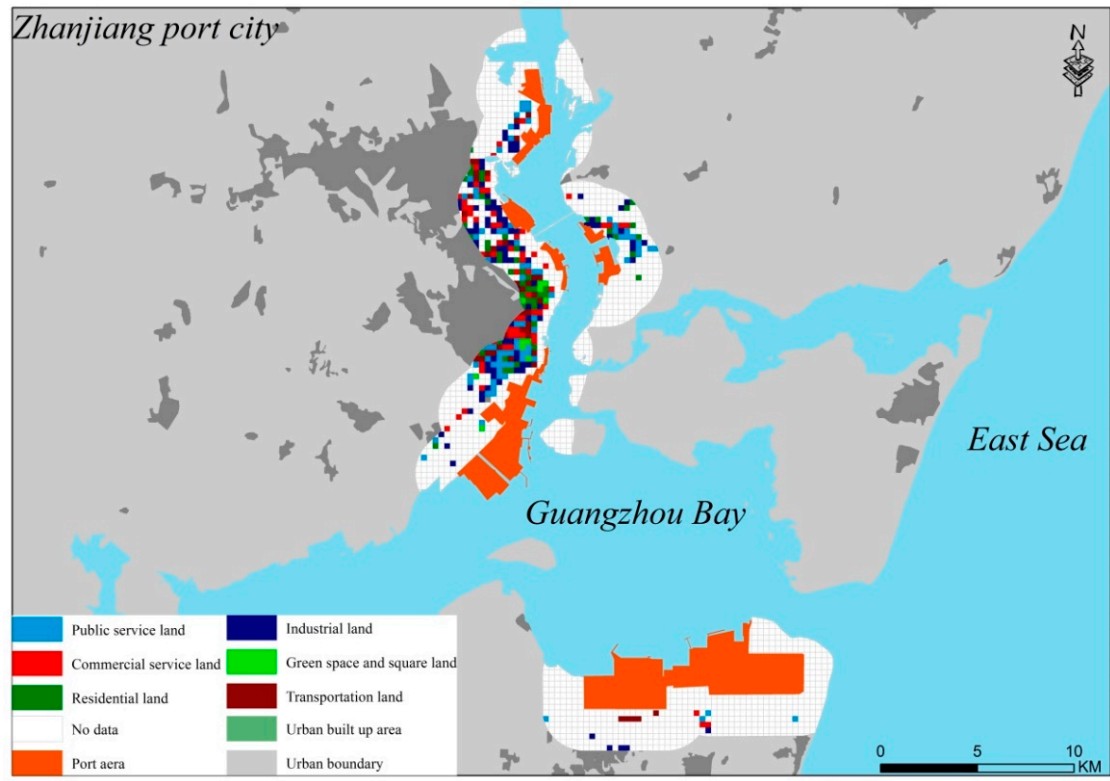

**Figure 12.** Distribution of urban functional areas on the Zhanjaing port–city interface (source: drawn by author).

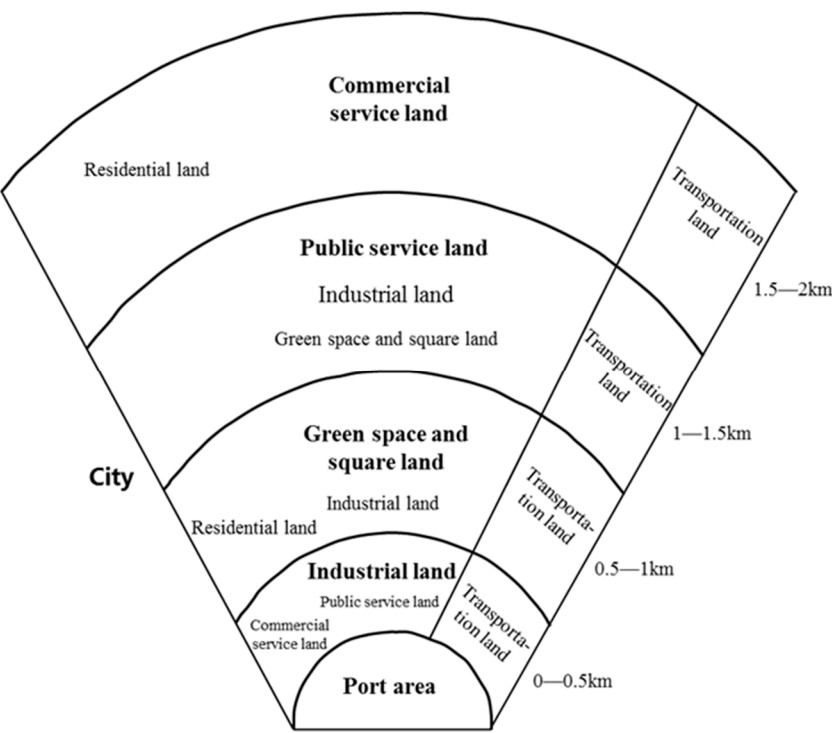

**Figure 13.** The urban regional structure model diagram of the coastal port–city interface (source: drawn by author).

## 7. Discussion and Conclusions

### 7.1. Discussion

First, this study, through the combination of remote sensing data and geospatial big data, identifies the spatial interaction between ports and cities from the perspective of landscape patches, which provides experience for the future spatial planning and management of ports and cities in ocean and coastal areas and complements the theories and methods of ocean and coastal management. According to our research, we believe that in coastal cities, ports are constantly driving urban growth, which is mainly reflected in the changes in the urban construction patches on the port–city interface and junction line. However, the spatial expansion of the port city will inevitably affect the ecological environment of the coastal zone, generating, for example, a biological habitat reduction caused by reclamation, frequent extreme weather, and maritime events, and even affecting human communities far away from the coastal zone [60]. Therefore, it is necessary to reasonably plan and construct the new port area to avoid excessive environmental impacts.

Second, starting from the port–city interface, this study defines the sequence of port-city spatial expansion, describes the spatial interaction and expansion direction of port cities, and analyzes the urban regional structure mode on the port–city interface. From the results, the average proportion of urban built-up areas formed before 1990 was 18.33%, and the average proportion of urban built-up areas formed between 1990 and 2020 was 81.68%. It can be seen that after the formation of ports, the expansion of urban built-up areas has been driven continuously. Although 37.76% of the port–city boundary has not yet formed urban built-up areas, it is still possible to form built-up areas in 2020 and beyond under the trend of port-city spatial integration in the future. However, in reality, the continuous global climate warming, rise of the water level, and the land-use change caused by the port and urban space expansion also have an important impact on the spatial relationship between ports and cities in coastal areas. On the one hand, the rise of water level may shrink the land area, increase the difficulty of selecting new port sites, determine the construction of deep-water ports far away from the city, and even force administrations to choose or reclaim islands. Therefore, the construction of the new port area increases the probability

of generating construction land in its periphery, and the driving role of the port in urban expansion is more obvious. On the other hand, the land change dominated by urban space expansion is present more inland, and the port space is expanding, resulting in the originally isolated port areas gradually connecting on the port–city interface, accelerating the driving role of the port in urban expansion.

Last, the combination of remote sensing data and geospatial big data provides new technologies and ideas for the research of port–city spatial relationship, which makes the port–city spatial relationship not limited to the economic dimension. The research paradigm of multiple data fusion can better summarize the spatial interaction relationship and mode of the port city in the future, judge the future development direction of the port city, and provide data and decision support for port-city spatial planning and transformation. It has important practical significance for the implementation of land and marine coordinated development strategy of China [43].

*7.2. Conclusions*

Taking the typical estuarine and coastal port cities in China's coastal zone as an example, this study combines remote sensing data with geospatial big data to identify the spatial interaction between ports and cities on the port–city interface. The main conclusions are as follows: First, the degree of spatial integration of port cities has gradually improved, but that of the estuarine port city is higher than the coastal port city, and the formation of ports has driven the expansion of urban built-up areas. Specifically, on the urban side of the port–city boundaries, an average of 62.24% have formed urban built-up areas, and 37.76% have not yet formed urban built-up areas. Among them, 31.17% of urban built-up areas were formed earlier than ports, and 68.83% of urban built-up areas were formed later than ports. Second, the expansion of ports has driven the expansion of urban built-up areas, and this expansion is port-oriented and sea-oriented, and the expansion direction of ports and cities is consistent. Specifically, the proportion of urban built-up areas formed before 1990 was only 17.87%, while the proportion of urban built-up areas formed between 1990 and 2020 was 82.13%. The proportion of urban built-up areas formed by coastal port cities before 1990 was only 18.78%, while the proportion of urban built-up areas formed between 1990 and 2020 was 81.22%. Third, different urban regional structure models are formed on the estuarine and coastal port–city interface. The urban regional structure mode of estuarine port city is: port area → industrial functional area → residential functional area → public service functional area → commercial service functional area. The urban regional structure mode of coastal port city is: port area → industrial functional area → green space and square functional area → public service functional area → commercial service functional area.

This study has several main contributions. First, from a micro-patch perspective, the sequence of port and urban spatial expansion on the port and city interface, the driving role of port expansion on urban expansion, and the spatial structure of the functional areas of the port's peripheral cities have been clarified. In addition, the research on the port and city relationship has been extended to the spatial level, while previous studies limited themselves to the industrial and economic relationship levels. Thus, it enriches the literature content on the relationship between port and city. Second, the results of this paper can provide reference for decision making departments to formulate a port-city development plan. Finally, regarding port cities, the expansion of ports in the past 30 years has significantly affected the expansion of cities, and the emergence of new port areas has led to the gradual integration of fragmented urban patches. However, the coastal port cities are located in the coastal zone area, which is a key area for the overall construction of land and sea in China. Therefore, the identification of the spatial interaction between ports and cities in the coastal area is conducive to the implementation of territorial space planning as well as to environmental protection and avoiding resource waste.

The evolution model of Western port cities shows that the development and evolution of port cities have gone through several centuries from the primitive stage to the modern

stage. Although China's port cities have a long history, they have undergone drastic changes in space starting in the 1980s [20]. The existing spatial evolution models of port–city relationships are mainly based on Western developed countries and cities, and there are only a few models in Asian developing countries and cities. The West takes a port in a country as an example, which is not universal. Compared with the Western model, due to the large number of port cities on China's coastal zone and the different location characteristics and scale of port cities, port cities are in different stages of development. Therefore, the single model cannot characterize the evolution of China's port cities. The development stage of the port city can be included in the research scope of port–city spatial relationship in the future research, and the evolution models of the port city in different stages can be summarized.

**Author Contributions:** W.L. contributed to all aspects of this work; Z.L. wrote the main manuscript text, conducted the experiment, and analyzed the data; Z.Z. and M.S. revised the paper. All authors reviewed the manuscript. All authors have read and agreed to the published version of the manuscript.

**Funding:** This research was funded by the Key Program of National Natural Science Foundation of China. National Natural Science Foundation of China, Grant number: 42030409.

**Data Availability Statement:** The collection and preprocessing of data are in Sections 3.1 and 3.2.

**Acknowledgments:** The authors would like to acknowledge all experts' contributions in the building of the model and the formulation of the strategies in this study.

**Conflicts of Interest:** The authors declare that they have no known competing financial interest or personal relationships that could have appeared to influence the work reported in this paper.

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
