# Peer review of "Research on the Interactive Relationship of Spatial Expansion between Estuarine and Coastal Port Cities"

_land, doi:10.3390/land12020371_

Round 1

Reviewer 1 Report

The authors present a possible combination of changing the space between the river estuary plateau and a plateau resulting from the ports' extension. 

The methodology is well defined; the literature review shows all problems and solutions concerning the analyzed issue's actual state of the art.

The authors have introduced their innovative procedure, confirmed by the results. Thus, the paper also has a practical aspect.

I am proposing a minor revision. Authors should give their vision about the impact of the water level rising, as well as changing the level of the ground. 

Author Response

请参阅附件。

Reviewer 2 Report

It was impprtant to indicate some relationshipd between ports and urban growth with correspondent envioronemental impacts.

Reviewer 3 Report

This manuscript is focused, well-written, and provides high-quality visuals. The authors are interested in analyzing the port-city spatial relationship, which is highly relevant to sustainable land development efforts. It has the potential to be considered for publication but needs to address some major issues.

My major concern with the manuscript is the methods used in this study are inadequate to support its claims. The authors need to more carefully conceptualize what they mean by "spatial interactions", or conduct more rigorous spatial analysis to provide sufficient evidence of the said spatial interactions.

Section 2, the literature review reads well overall. I would like to see the authors more clearly articulate the gaps in the literature and this study fits into the literature stream. There are some good discussions in section 7.1, I suggest move them here to justify why quantitative research like this is needed. 

Section 3, the authors stated that they performed remote sensing analysis, but it seems the data they used are vector layers provided by third-party institutions, rather than the original remote sensing images. Please provide more information on this vector data – what attributes does it contain, how the land uses are classified? Please also clarify the work the authors have done – did you process the remote sensing images or just calculated the statistics of the processed data?

3.2, Line 183, “the average building area or floor area of various POIs is roughly determined”. This work is not trivial – please detail your processes, what auxiliary data have been used to determine the building areas and floor areas, how did you decide the building heights, what assumptions have been for mixed-use buildings, etc.

3.2, Line 185-191, I’m not sure I understand what the weight score is used for here. Why do we need to scale up the land use areas?

3.3.1, Line 210, what is the “graphic induction” method? This needs more information or citations from previous studies.

3.3.1 Line 212, How are the qualities of these images, which months are selected, how many images are used for each year?

Section 4, the authors evaluate and make conclusions about 1) “the degree of port-city spatial integration” and 2) whether the port is the driving force for the formation of the city, based on three metrics, 1) the proportion of built-up areas, 2) length of port-city boundaries and 3) the built-up areas patch length. Please provide more explanations to your rationale because I am not convinced the metrics provide sufficient evidence for the conclusions. First, the concept of “spatial integration” is vague, does it mean a higher percentage of urbanized areas? Second, I don’t think simple proportions and lengths lend themselves to a generalized conclusion of spatial development patterns, given the many different shapes, development trajectories, and levels of development among these different ports. Third, the causal relationship of “which drives which” cannot be simply achieved from the area/length measurements. The authors are making a lot of assumptions. Their conclusions need to follow the results to a larger degree.

Table 1, there is a typo “Tapy” on the 1st and 7th columns. 

Table 1, the “proportion” on the 6th and last columns are not self-explanatory. More text annotations are suggested.

Section 6, the author developed two metrics “frequency density” and “category ratio” in 3.3.2, and states they “select the POI type with the highest proportion as the main land 395 type of the research unit”. Please clarify the corresponding values in Table 4 - which values refer to F and which refer to C? The columns names “Quantity” and “Proportion” need to be explained.

Figure 6-8, why are some cells “no data” and what does that mean?

Figure 9 and 10. I find these two diagrams disconnected from the POI maps shown above. I do not see a clear structure as claimed in Figure 6-8.

Section 6.1 The POIs used are from 2020. How does this “snapshot” of very recent urban amenities reflect the region’s development patterns over time?

Conclusions, the authors can be more explicit about their contributions. Their findings including the amount of built-up areas and the derived spatial structures are nicely summarized, but a statement of contribution is needed here as of why we should know these results and how they add to our understanding about port-city relationship.

Round 2

Reviewer 3 Report

Thanks for the revision.